# Transcriptome Dynamics Underlying Planticine^®^-Induced Defense Responses of Tomato (*Solanum lycopersicum* L.) to Biotic Stresses

**DOI:** 10.3390/ijms24076494

**Published:** 2023-03-30

**Authors:** Roksana Rakoczy-Lelek, Małgorzata Czernicka, Magdalena Ptaszek, Anna Jarecka-Boncela, Ewa M. Furmanczyk, Kinga Kęska-Izworska, Marlena Grzanka, Łukasz Skoczylas, Nikodem Kuźnik, Sylwester Smoleń, Alicja Macko-Podgórni, Klaudia Gąska, Aneta Chałańska, Krzysztof Ambroziak, Hubert Kardasz

**Affiliations:** 1INTERMAG Sp. z o.o., Al. 1000-lecia 15G, 32-300 Olkusz, Poland; 2Department of Plant Biology and Biotechnology, Faculty of Biotechnology and Horticulture, University of Agriculture in Krakow, Al. Mickiewicza 21, 31-120 Krakow, Poland; 3Department of Plant Protection, The National Institute of Horticultural Research, Konstytucji 3 Maja 1/3 Str., 96-100 Skierniewice, Poland; 4Department of Plant Product Technology and Nutrition Hygiene, Faculty of Food Technology, University of Agriculture in Krakow, Balicka 122 Str., 30-149 Krakow, Poland; 5Faculty of Chemistry, Silesian University of Technology, Krzywoustego 4 Str., 44-100 Gliwice, Poland; 6Laboratory of Mass Spectrometry, Faculty of Biotechnology and Horticulture, University of Agriculture in Krakow, Al. 29 Listopada 54, 31-425 Krakow, Poland; 7NEFSCIENCE, Bohaterów Westerplatte 119 Str., 96-100 Skierniewice, Poland

**Keywords:** elicitor, oligogalacturonides, plant-pathogen interaction, phytohormones, RNA-seq, transcriptome

## Abstract

The induction of natural defense mechanisms in plants is considered to be one of the most important strategies used in integrated pest management (IPM). Plant immune inducers could reduce the use of chemicals for plant protection and their harmful impacts on the environment. Planticine^®^ is a natural plant defense biostimulant based on oligomers of α(1→4)-linked D-galacturonic acids, which are biodegradable and nontoxic. The aim of this study was to define the molecular basis of Planticine’s biological activity and the efficacy of its use as a natural plant resistance inducer in greenhouse conditions. Three independent experiments with foliar application of Planticine^®^ were carried out. The first experiment in a climatic chamber (control environment, no pest pressure) subjected the leaves to RNA-seq analysis, and the second and third experiments in greenhouse conditions focused on efficacy after a pest infestation. The result was the RNA sequencing of six transcriptome libraries of tomatoes treated with Planticine^®^ and untreated plants; a total of 3089 genes were found to be differentially expressed genes (DEGs); among them, 1760 and 1329 were up-regulated and down-regulated, respectively. DEG analysis indicated its involvement in such metabolic pathways and processes as plant-pathogen interaction, plant hormone signal transduction, MAPK signaling pathway, photosynthesis, and regulation of transcription. We detected up-regulated gene-encoded elicitor and effector recognition receptors (ELRR and ERR), mitogen-activated protein kinase (MAPKs) genes, and transcription factors (TFs), i.e., *WRKY*, *ERF*, *MYB*, *NAC*, *bZIP*, pathogenesis-related proteins (PRPs), and resistance-related metabolite (RRMs) genes. In the greenhouse trials, foliar application of Planticine^®^ proved to be effective in reducing the infestation of tomato leaves by the biotrophic pathogen powdery mildew and in reducing feeding by thrips, which are insect herbivores. Prophylactic and intervention use of Planticine^®^ at low infestation levels allows the activation of plant defense mechanisms.

## 1. Introduction

The European market for plant protection products is regulated by Regulation (EC) 1107/2009, which defines the criteria for pesticide approval for use, complemented by Directive 2009/128/EC, which requires member states to develop a policy of sustainable use for pesticides. One of the aims of this directive is to encourage the development and introduction of integrated pest management (IPM) in order to reduce the use of pesticides in agricultural practice. Activation of natural defense mechanisms in plants is considered to be one of the most important strategies used in IPM. Plant resistance stimulants are a class of compounds that increase and strengthen the natural resistance of plants. The efficacy of such stimulants is almost as effective as pesticides, which could reduce the use of chemical compounds and thus their adverse effect on the environment, humans, and pollinators [1,2].

Plants do not have an as advanced immune system as animals but are able to show resistance to harmful organisms and the damage they cause. Innate immunity in the form of a physical and chemical barrier is present in the plant throughout its life and is divided into non-specific resistance, which provides various plants with an effective defense against various species and strains of pathogens, and specific resistance, which determines the protection of a specific, single type of plant against one or more pathogenic strains. Plant-acquired immunity is triggered in response to an attack by pathogens and pests. It is activated in cells surrounding the site of infection (locally acquired immunity) or develops later in remote parts of plants as Systemic Acquired Resistance (SAR) and Induced Systemic Resistance (ISR) [3,4,5,6]. SAR is trigged by localized pathogen attacks and uses the salicylic acid (SA) pathway to transduce the signal in the whole plant. ISR is triggered by nonpathogenic and plant growth-promoting microorganisms, including fungi (PGPFs) or rhizobacteria (PGPRs). ISR relies on jasmonic acid (JA) and ethylene (ET) to transduce the defense signal in the whole plant [7,8,9].

Pathogens enter the plant (host) tissue by direct penetration of the plant surface, through physical injuries, or through natural structures such as stomata. Activation of the plant’s defense response begins on the surface of its aerial parts when the harmful organism induces changes to the cuticle, which plants recognize [6,10]. The first line of plant defense is the cell wall, which constitutes a physical barrier between pathogens and the internal content of cells. This consists of complex polysaccharides and is covered by a layer of wax which determines its defense properties. Pathogens produce hydrolytic enzymes that decompose the cell wall to access nutrients contained in the host cell [11,12,13]. The second line of defense is a chemical barrier (production of antimicrobial compounds). Plants recognize pathogens and insects through their secretomes and other molecular patterns, which interact with the plant cell surfaces and induce plant signal molecules that activate signal transduction cascades and then defense and resistance genes in plants [6,14,15].

The molecule patterns produced by pathogens and insects during infection and insect feeding are pathogen-associated molecular patterns (PAMPs), microbe-associated molecular patterns (MAMPs), herbivore-associated molecular patterns (HAMPs), and damage-associated molecular patterns (DAMPs). These molecular patterns are non-specific elicitors, which at the molecular level, result in the activation of pattern-triggered immunity (PTI) [4,6,15,16,17,18].

Activation of PTI causes, among other things, alkalization of the cytoplasm due to a large influx of calcium ions, production of reactive oxygen and nitrogen species, and the activation of mitogen-activated protein kinases (MAPK) [18,19]. MAPKs cause the activation of transcription factors affecting the expression of pathogenesis-related (PR) genes, the production of ethylene, JA, and SA, the strengthening of the plant cell wall, and the induction of the synthesis of antimicrobial compounds [20].

Elicitors can be substances of natural origin isolated from crustaceous or algae materials such as chitosan or hepta-β-glucoside that can imitate the plant-pathogen interaction and induce defense mechanisms in plants by binding receptor molecules found in the plant plasma membrane [6,21,22]. It has been shown that chitosan induced antifungal mechanisms in horticultural crops such as carrots, cucumbers, and tomatoes [23], while hepta-β-glucoside induced synthesis of phytoalexin production in white lupin, alfalfa, beans, and potatoes [24].

Oligogalacturonides (OGs) are also well-known elicitors that have been widely tested as plant growth biostimulants and inducers of plant defense. OGs are fragments of pectin, a main constituent of the plant cell wall, and belong to the class of oligosaccharides [13,25,26].

Planticine^®^, created by INTERMAG Sp. z o.o. (Olkusz, Poland), is a unique natural biostimulant of plant defense mechanisms that is a mixture of oligomers of α(1→4)-linked D-galacturonic acids with a degree of polymerization (DP) from 2 to 10. Planticine^®^ is a biodegradable, non-toxic, and water-soluble substance, which makes it attractive for applications in agriculture. Planticine^®^, used both prophylactically and interventionally, effectively reduces infestation of plants by pathogens and pest feeding.

The aim of this study was to determine the efficacy of Planticine^®^ in reducing damage caused by agrophages and to determine the mode of action of this biostimulant. Transcriptome analysis of tomatoes treated with Planticine^®^ was used to define the molecular basis of the biological activity of Planticine^®^.

## 2. Results

### 2.1. Genome-Wide Identification of Expressed Genes in Tomatoes Exposed to Planticine^®^

The six transcriptome libraries of tomatoes subjected to Planticine^®^ treatment and control plants were profiled using Illumina paired-end (PE) 2 × 150 bp sequencing. A total of 289 million (M) reads (40.8 Gbp), with an average of 48.2 M reads for each library, were produced with a Q30 quality score (sequencing error rates < 0.1%) equal to 96% (Appendix A). A total of 96% of the reads were mapped to the *S. lycopersicum* reference genome (GCA_000188115.3), and about 84% of reads were uniquely mapped to genes (Appendix A). In addition, the Pearson correlation analysis between three biological replicates was greater than 0.98, indicating the reliability of the RNA-seq results (Appendix A). A total of 24,154 expressed genes were identified in this study, among which 23,977 and 23,975 were in Planticine^®^-treated plants and control plants, respectively (Appendix A).

A total of 3089 genes were found to be differentially expressed after exposure to Planticine^®^, among which 1760 were up-regulated while 1329 were down-regulated (Figure 1A, Appendix A).

To better elucidate the biological functions of the DEGs, a Kyoto Encyclopedia of Genes and Genomes (KEGG) pathway analysis was conducted. In total, 160 DEGs were significantly enriched in 7 tomato pathways, i.e., plant hormone signal transduction (sly04075), plant-pathogen interaction (sly04626), MAPK signaling pathway-plant (sly04016), photosynthesis (sly00195), porphyrin and chlorophyll metabolism (sly00860), photosynthesis-antenna proteins (sly00196), and carotenoid biosynthesis (sly00906) (Figure 1B, Appendix A).

The identified DEGs were also functionally annotated with Gene Ontology (GO) terms and classified as a biological process (BP), molecular function (MF), and cellular component (CC). In total, 2331 (75.5%) DEGs were annotated with GO terms, among which 1561 DEGs were assigned to 107 BP terms, 1911 DEGs to 49 MF terms, and 1285 DEGs to 24 CC terms (Figure 1C, Appendix A). These genes were significantly enriched (*p*-values ≤ 0.01) in biological processes, such as defense response to other organisms (GO:0098542), protein phosphorylation (GO:0006468), regulation of transcription, DNA-templated (GO:0006355), photosynthesis, light harvesting in photosystem I (GO:0009768), phototropism (GO:0009638), response to auxin (GO:0009733), regulation of the JA-mediated signaling pathway (GO:2000022), and regulation of the SA biosynthetic process (GO:0080142) (Figure 1C). Top molecular function annotations included protein kinase activity (GO:0004672), DNA binding transcription factor activity (GO:0003700), chlorophyll binding (GO:0016168), calcium ion binding (GO:0005509), and iron ion binding (GO:0005506) (Figure 1C). All DEGs assigned to the chlorophyll binding term were up-regulated (Appendix A). Furthermore, many DEGs were assigned to cellular component terms, such as an integral component of membrane (GO:0016021), photosystem II (GO:0009523) and photosystem I (GO:0009522), cell wall (GO:0005618), extracellular region (GO:0005576), and SCF ubiquitin ligase complex (GO:0019005) (Figure 1C, Appendix A).

These results imply that most of the KEGG and the GO assignments of identified DEGs were plant-pathogen interaction, plant hormone signal transduction, and photosynthesis-responsive genes associated with Planticine^®^ treatment.

#### 2.1.1. Genes Related to Plant-Pathogen Interaction

Resistance in plants against pathogens is known to be controlled by a hierarchy of genes, i.e., elicitor and effector recognition receptors (ELRR and ERR), mitogen-activated protein kinases (MAPKs), transcription factors (TFs) and other regulatory genes, phytohormones that finally lead to biosynthesis of resistance-related proteins (RRPs), and metabolites (RRMs) that directly suppress the pathogen.

Among the 3089 DEGs, we detected 18 up-regulated gene-encoded elicitor and effector recognition receptors (ELRR and ERR), including receptor-like proteins EIX1 and EIX2, chitin elicitor receptor kinase 1, LysM domain receptor-like kinase 4 (CERK1/*LYK4*), receptor-like protein kinase (*RLK*), probable cyclic nucleotide-gated ion channel (*CNGC*), flagellum-specific ATP synthase (*FLII*), LRR receptor-like serine/threonine-protein kinase (*SIK1*, *GSO2*, *RFK1*, *FEI2*), wall-associated receptor kinase-like (*WAK*), leaf rust 10 disease-resistance locus receptor-like protein kinase-like (*LRL*), and somatic embryogenesis receptor kinase 1 (*SERK1*). Only three DEGs, i.e., elongation factor (*ELF*), chitin elicitor receptor kinase 1, LysM domain receptor-like kinase 4 (CERK1/*LYK3*), and the LRR receptor-like serine/threonine-protein kinase FLS2 were down-regulated in the Planticine^®^-treated samples (Appendix A, Table 1).

Several mitogen-activated protein kinases (MAPKs) genes, which were activated by receptor genes, were found to be up-regulated under Planticine^®^ treatment, i.e., *SIMAPKK76*, *CDPK*, *CML*, and *MPK3* (Figure 2, Table 1, Appendix A).

Transcriptional control of the expression of resistance genes plays a crucial role in response to stress in plants. In our research, *WRKY*, *ERF*, *MYB*, *NAC*, and *bZIP* were five types of transcription factors (TFs) detected. Among 19 genes encoding the WRKY TFs, only two were down-regulated (*WRKY 19*, *WRKY21*). Ten genes encoding ERF TFs were detected with increased transcript accumulation in the Planticine^®^-treated plants, while two were down-regulated (*ERF3*, *ERF13*). The 7 MYB TFs, 4 NAC-related, and 1 BZIP detected were induced in the Planticine^®^-treated tomatoes. Interestingly, circular RNA (circRNA) whitefly-induced gp91 was up-regulated (Appendix A, Table 1), and CircRNAs were proved to act as miRNA sponges and inhibit miRNA activity.

Both pathogenesis-related proteins (PRPs) and resistance-related metabolites (RRMs) directly suppress pathogens. Planticine^®^ triggered overexpression of PRP genes, such as A70 (pathogen-associated molecular patterns-induced protein A70), *PTI5*, *PTI6* (pathogenesis-related gene transcriptional activator PTI5 and PTI6), *PRR*, *PRS2* (pathogenesis-related protein), *RIN1*, *RIN4* (RPM1-interacting protein 1 and 4), *R13L4*, *ROQ1*, *RPV1*, *RUN1*, *NGR1*, *EDR1* (disease resistance protein), *HSP12*, *HSP71* (heat shock protein), *PBS1* (probable serine/threonine-protein kinase), *EDS1* (protein EDS1), *KCS11* (3-ketoacyl-CoA synthase 11), *HIR1* (hypersensitive-induced reaction 1 protein), and *MCA1* (metacaspase-1) (Appendix A, Table 1).

All detected resistance-related metabolite (RRMs) genes were accumulated to higher transcript levels in tomatoes subjected to Planticine^®^ treatment. Four genes related to JA metabolism were overexpressed, i.e., encoded linoleate lipoxygenase (*LOX31*, *LOX21,* and *LOX15*), allene oxide synthase (*AOS*), (R)-linalool synthase TPS5 (*TPS5*), and 1-aminocyclopropane-1-carboxylate synthase (*ACS*) (Appendix A, Table 1).

#### 2.1.2. Genes Related to Plant Hormone Signal Transduction

Planticine^®^ treatment alters the expression of genes related to the phytohormone signal transduction pathway (Appendix A).

For the SA biosynthetic pathway, it was possible to detect two DEGs encoding phenylalanine ammonia-lyase (*PAL5*). We were able to detect six DEGs associated with the SA signal transduction pathway encoding lipase-like PAD4 (*PAD4*), calmodulin-binding protein (*CBP*), negative protein regulator of resistance (*NPR1*), and pathogenesis-related protein (*PRR*). Application of Planticine^®^ caused up-regulation of all identified DEGs (Figure 2A).

We observed changes in the activity of genes related to JA (Figure 2B). We were able to detect nine DEGs, all up-regulated, associated with the jasmonate signal transduction pathway encoding protein TIFY (*TIF5A*, *TI10A*, *TI10B*, *TIF3B*), jasmonate ZIM domain-containing protein (*JAZ2*, *JAZ7*), BTB/POZ domain and ankyrin repeat-containing protein (*NPR1*), and transcription factors, i.e., MYC2 and ERF1. In addition, we were able to find DEGs directly related to α-linolenic acid metabolism leading to jasmonate production. We identified genes encoding key enzymes involved in this pathway, such as lipase (*LIP*), lipoxygenase (*LOX1*, *LOX15*, *LOX21*, *LOX31*), allene oxide synthase 1 (*AOS1*), and 12-oxophytodienoate reductase 1 (*OPR1*). Among these, DEGs only *LIP*, *LOX1,* and *OPR1* were down-regulated.

We next analyzed the genes proposed to be involved in the abscisic acid (ABA) pathways (Figure 2C). A total of sixteen DEGs related to biosynthesis, metabolic process, and signal transduction pathway were identified. The majority of genes related to biosynthesis were down-regulated. However, the expression of four genes involved in the ABA metabolic process was up-regulated, i.e., nodulin-related protein 1 (*NDRP1*), 9-cis-epoxycarotenoid dioxygenase (*NCED1* and *NCED2*), and abscisic acid 8′-hydroxylase (*ABAH2*). The expression of genes involved in ABA signal transduction was down-regulated for the ABA receptor (*PYL10*) and ethylene-rSerine/threonine-protein kinase SRK2C (*SRK2C*) and up-regulated for protein phosphatase 2C 2, 63, and 77 (*PP2C2*, *PP2C63,* and *PP2C77*).

### 2.2. The Content of Phytohormones

Foliar application of Planticine^®^ at a dose of 2 L ha^−1^ in the cultivation of plants growing under controlled conditions had a statistically significant effect on the content of phytohormones in the tomato leaves. Foliar application of Planticine^®^ caused a significant increase in the content of SA with a significant simultaneous decrease in the content of JA in the leaves compared to the control. ABA content after the application of Planticine^®^ was not statistically significantly different compared to the control treatment (Table 2).

### 2.3. Powdery Mildew on Tomato Leaves

Planticine^®^ and chemical standard were applied preventively before the outbreak of the disease occurred. The first symptoms of powdery mildew on the tomatoes were observed after the third spraying of the plants. The tested products in each of the three observations showed an inhibitory effect on powdery mildew development, as was confirmed by statistical analysis. The chemical standard Scorpion 325 SC significantly reduced the infestation of plants by *O. neolycopersici* compared to the control and Planticine^®^ (Table 3). The chemical standard was the most effective in protecting plants against powdery mildew. In the combination where Scorpion 325 SC was applied, no disease symptoms were observed (efficacy 100%) until 10 days after the last spraying. The efficacy of the chemical standard was then calculated to be 99%. Planticine^®^ showed a significant increase in the efficacy for the dose of 2 l ha^−1^ during each observation compared to the dose of 3 l ha^−1^. However, it should be emphasized that the dose of 3 l ha^−1^ also reduced disease development. The level of leaf infestation on the first observation date (T3 + 10) was respectively 8.8% in the control combination, about 1% for Planticine^®^ treatment (both doses), and 0% for Scorpion 325 SC. The degree of leaf infestation by the pathogen after Planticine^®^ treatment was significantly lower than in the control combination. The efficacy of both Planticine^®^ tested doses was on a comparable level of 90–91%. On the second date of observation (T4 + 10), the level of leaf infestation in the control combination was 30.2%, whereas, in combinations treated with Planticine^®^, this was 7.4% and 10.4% depending on the dose and 0% after Scorpion 325 SC treatment. Significant statistical differences were observed in the degree of tomato tissue infestation by powdery mildew depending on the dose of Planticine^®^. The dose of 2 l ha^−1^ was more effective in protecting tomatoes against pathogens than the dose of 3 l ha^−1^.

### 2.4. Thrips on Tomato Leaves

Planticine^®^ and chemical standards were applied interventionally. The observations which were carried out 3 days after the first application (T1 + 3) showed that only Planticine^®^ applied at the dose of 2 l ha^−1^ reduced the number of adult thrips with an efficacy of 45% (Table 4). This tendency was observed on the next two observation dates. Repeated application increased the efficacy of Planticine^®^ up to 61% 7 days after the 2nd treatment and 74% 14 days after the 2nd treatment. A higher dose of Planticine^®^, except for the last observation date (T2 + 21), did not reduce the number of adult thrips. The application of Mospilan 20 SP did not cause a significant decrease in the number of thrips during the first four observations (Table 4).

## 3. Discussion

Oligogalacturonides elicit diverse biological effects in plants. There are well-known examples of stimulation of molecular and physiological processes by OGs of DP 9–15, including growth promotion [27,28], synthesis of antioxidant enzymes [29], activation of defense responses [30], expression of genes encoding pathogenesis-related proteins [31], and accumulation of phytoalexins [32,33]. Nevertheless, studies involving OGs of DP < 7 have shown that short fragments also exhibit biological activity. Dimeric OGs activated proteinase inhibitor synthesis in tomato seedlings [34]; di- and trimeric ones induced plant defense response against pathogens [35,36]. OGs of DP 1–7 do not show differences in mode of action compared to fragments with a higher DP degree in the range of 10–20 [31]. Analysis of transcriptional profiling in Arabidopsis thaliana seedlings conducted by Denoux et al., 2008 [29] showed that there were no significant differences in the activation of plant defense response between short and long fragments of OGs. The results we present here for Planticine^®^ confirm that there is not necessarily any minimum DP limit for OG activity. Planticine^®^, which contains OGs with a polymerization degree from 2 to 10, activates the natural defense mechanisms of plants, increasing their resistance to agrophage infection.

The elicitors of PAMPs, MAMPs, HAMPs, and DAMPs are produced by pathogens or are formed as a result of damage to plant tissues during pathogen infection, pest feeding, and as a consequence of mechanical injury. Elicitors are recognized by membrane receptors and activate plant-pathogen interaction, in which the three stages of signal perception, signal transduction, and defense response can be distinguished [6,14,37,38,39].

The cDNA analysis showed that expression of genes related to plant-pathogen interaction at each of the three stages of signal perception, signal transduction, and defense response was increased in tomato plants treated with Planticine^®^ that were not exposed to biotic stresses. Planticine^®^ mimics elicitors acting as PAMPs, MAMPs, HAMPs, and DAMPs and triggers the first stage of the plant-pathogen interaction, i.e., signal perception. This stage starts when the plasma membrane proteins recognize the elicitor. Planticine^®^ increased expression of the genes belonging to the class of elicitor recognition receptor genes (ELRR), such as chitin elicitor receptor kinase 1, LysM domain receptor-like kinase 4 (*CERK1/LYK4*), and receptor-like protein kinase (*RLK*). It is known that these genes are activated by substances produced by hemibiotrophs and necrotrophs [37,38,39]. Planticine^®^ also increased expression of the gene encoding wall-associated receptor kinase-like (*WAK*), which is activated by substances produced by necrotrophs and substances formed from the damage of the cell wall both from mechanical injuries and those caused by herbivorous insects [6]. Plants treated with Planticine^®^ exhibited increased expression of the gene encoding the leucine-rich repeats (LRR)-containing domain, which is part of many proteins associated with innate immunity in plants, i.e., NBS (nucleotide-binding site). LRR proteins are plasma membrane proteins that recognize elicitors produced by biotrophs and some necrotrophs [37,38,40].

In the signal transduction stage, the main role is played by the MAPK kinase kinase (MAPKKK) pathway, which receives the signal from plasma membrane proteins and transmits it through cytosolic kinases to the nucleus to activate transcriptional factors and defense-related genes for SAR [6,14,39]. Planticine^®^ activated a number of mitogen-activated protein kinase (MAPKs) genes which encoded kinases that act as signal transducers in the MAPKK pathway.

In the defense response, which is the last stage of plant-pathogen interaction, Planticine^®^ increased the expression of transcription factors (TFs), i.e., *WRKY*, *ERF*, *MYB*, *NAC*, which regulate the expression of plant disease resistance genes (*R* genes) to produce pathogenesis-related proteins (PRPs) and resistance-related metabolites (RRMs) [39]. Besides transcription factors, the expression of genes encoding PRPs and RRMs was also enhanced by Planticine^®^. The PRPs, also called PR proteins, are a structurally diverse group of plant proteins that show strong antifungal and antimicrobial activity. PR proteins are either extremely acidic or extremely basic and therefore are very reactive [41]. On the other hand, RRMs include phytoalexins and phytoanticipins or products of their conjugate that are deposited to enforce the secondary cell wall, thus containing the pathogen in the initial infection area [14,42,43]. Phytoalexins are toxic mostly to pathogenic fungi but also to bacteria and nematodes [41]. Furthermore, among the up-regulated genes, we identified the *WF11* gene, which was annotated as whitefly-induced gp91—circular RNA. Hong et al., 2020 [44] first identified circRNAs in tomatoes experiencing *Phytophthora infestans* infection and demonstrated that whitefly-induced gp91 might act as a positive regulator in tomato resistance by regulating miRNA-mRNAs expression levels.

The presented analysis of gene expression related to the plant-pathogen interaction in healthy tomato plants treated with Planticine^®^ not exposed to biotic stresses caused by agrophages allows us to conclude that Planticine^®^ mimics the pathogen-plant and/or insect-plant interaction and acts as an elicitor produced by biotrophs, hemibiotrophs, and necrotrophs. Planticine^®^ is an elicitor that activates plant immune reactions, including SAR. Planticine^®^-activated genes related to plant hormone biosynthesis and signal transduction engaged in the activation and development of SAR. SA is responsible for the activation of SAR and the production of PR proteins [6,39,41,45]. Biosynthesis of this phytohormone occurs via the shikimic acid pathway, which forms two distinct branches, both of which synthesize SA. The first involves isochorismate synthase (ICS), and the second involves phenylalanine ammonia-lyase (PAL) [45,46,47,48]. Planticine^®^ influenced the synthesis of SA by increasing the expression of the gene encoding the PAL enzyme, which directly resulted in a significant increase in the concentration of SA in the leaves of tomatoes treated with the tested product. Planticine^®^ not only activated genes responsible for SA synthesis and thus indirectly SAR induction but also increased the expression of further genes important for SAR, i.e., nonexpressor of pathogenesis-related genes 1 (*NPR1*) and pathogenesis-related (*PR*) genes. NPR1 genes encode NPR1-like proteins, which are transcription factors that play a significant role in the establishment and development of SAR [45,49]. Among PR genes encoding PR proteins, well-known examples are PR1 proteins (antioomycete and antifungal), PR2 (b-1,3-glucanases), PR3 (chitinases), PR4 proteins (antifungal), PR6 (proteinase inhibitors), thaumatine-like proteins, defensins, thionins, lysozymes, osmotin-like proteins, lipoxygenases, cysteine-rich proteins, glycine-rich proteins, proteinases, chitosanases, and peroxidases [41]. This study showed that Planticine^®^ increased the expression of the PR1 gene, which encodes proteins with antifungal properties.

Planticine^®^ influenced the activation of genes related to JA, which is the second important plant hormone also responsible for the plant’s response to pathogens and the induction of ISR. In the JA-dependent signal transduction pathway, of particular concern is the *NPR1* gene, whose expression was increased in both the pathway for SA and JA. However, *NPR1* gene expression in the SA acid pathway was 3.5 times higher than in the JA pathway. This is related to another important role played by NPR1-like proteins, which is mediating crosstalk between the SA and JA responses [39,45]. In the presented study, crosstalk between SA and JA has an antagonistic character, which was confirmed by the expression level and analysis of the content of both hormones in tomato leaves. The significant increase in SA was accompanied by a significant decrease in JA. These results are confirmed by the work of other authors, who observed a negative interaction between the JA and SA pathways [50,51,52].

Abscisic acid is a phytohormone involved in the regulation of plant growth and development, which is synthesized as a result of abiotic stress and is important in the processes of plant acclimatization to changing environmental conditions [53]. Planticine^®^ mainly activated plant response pathways to biotic stress without stimulating abiotic stress response pathways. ABA biosynthesis genes were mostly down-regulated, while ABA content in the leaves of tomatoes treated with Planticine^®^ was not significantly different from leaves in the control. The lack of effect of Planticine^®^ on abscisic acid synthesis indirectly indicates that the application of the Planticine^®^ formulation alone is not harmful to plants.

To determine the efficacy of the tested product in stimulating plant defense processes and increasing plant resistance to pathogen and pest attacks, two independent greenhouse experiments were conducted with the Planticine^®^ application in tomato cultivation. The Planticine^®^ used prophylactically showed a high efficacy of approx. 90% in reducing tomato powdery mildew (*O. neolycopersici*) in the initial stage of disease development. The increase in pathogen pressure observed on the 2nd and 3rd assessment dates, reflected in control plants by infestation at the level of 30% of infected leaf tissues, followed by more than 60% infection of the leaf area, resulted in a reduction of the efficacy to the level of 76% for the dose of 2 l ha^−1^ (20 g OGs ha^−1^) and 66% for 3 l ha^−1^ (30 g OGs ha^−1^) during the second observation and 47% and 38%, respectively, during the third observation. The high efficacy of Planticine^®^, which remained at the level of 90–76/66% for the first two assessments, indicates that Planticine^®^ used prophylactically acted as an elicitor, activating the defense processes of tomato plants and increasing their resistance to powdery mildew infestation. The application of the Planticine^®^ did not completely eliminate powdery mildew, as was the case with the chemical reference product; however, it significantly reduced the pathogen infestation of the plants. Planticine^®^ was also prophylactically used in greenhouse cucumber cultivation. Planticine^®^ at a dose of 2 l ha^−1^ allowed a significant reduction in the occurrence of cucumber powdery mildew (*Golovinomyces orontii*) on leaves at efficacy levels of 60% and 50% compared to the untreated control (own unpublished data). Similar results were obtained by Aubel et al. [1], who studied the efficacy of an alternative elicitor formulation containing a complex of oligochitosans and oligopectates (COS-OGA) against cucumber powdery mildew (*Sphaerotheca fuliginea*) in greenhouse conditions. The efficacy of COS-OGA at a spraying rate of 25 g ha^−1^ caused approximately a 70% reduction in leaf disease severity.

Interventional testing of Planticine^®^ in greenhouse tomato cultivation against thrips (*F. occidentalis*) showed varied efficacy in reducing the numbers of adult thrips feeding on tomato leaves. The first application showed efficacy at the level of 45% and 20%; however, repeating the treatment increased the efficacy of controlling adult thrips to 61% and 74%. The highest efficacy against thrips was observed 14 days after the 2nd application. Apart from the last observation, the efficacy of Planticine^®^ against thrips was higher than the chemical reference product. The efficacy of this biostimulant increased after some time after application, which is a strong confirmation that Planticine^®^ acted as an elicitor and can also be used as interventional application. The interesting fact is that the thrips did not feed as much on the plants sprayed with Planticine^®^ as they did on the other combinations, including the insecticide combination. These results confirm that Planticine^®^ activation of PRPs and RRMs genes results in increased tissue concentrations of secondary metabolites that inhibit herbivorous insect digestion. The chemical structures of phytoalexins belonging to the class of RRMs produced by plants in the Solanaceae family are terpenoids [41].

Foliar application of Planticine^®^ proved to be effective in reducing the infestation of tomato leaves by the biotrophic pathogens powdery mildew and in reducing feeding thrips belonging to the order Thysanoptera, which are herbivorous insects. Prophylactic and interventional use of Planticine^®^ at low infestation level allows activation of plant-pathogen interaction pathway genes, defense-related genes of SAR, and accumulation of PR proteins and RRMs.

Both in the experiment with the fungal pathogen and pests, treatment of tomatoes with Planticine^®^ did not inhibit the completion of the life cycle of *O. neolycopersici* or *F. occidentalis*, but it decreased the progression of infestation by powdery mildew and feeding thrips. This resulted in a reduction in the leaf area covered by symptoms of disease and feeding thrips in the experiments. Although this result does not indicate that the use of Planticine^®^ should replace synthetic fungicides or insecticides, this biostimulant of plant defense may still be useful in combination with other control strategies in IPM programs based on reduced pesticide use. The efficacy of disease and pest control by using such products containing plant extracts, natural substances, or living organisms and their metabolites may not be as high as that of synthetic chemical plant protection products [54], which was shown in this research.

## 4. Materials and Methods

### 4.1. Tomato Trail in the Climatic Chamber

To determine the molecular basis of the biological activity of Planticine^®^, the product was applied to tomatoes grown under controlled conditions in a climatic chamber (property of INTERMAG Sp. z o.o., Olkusz, Poland). Planticine^®^ containing 10 g L^−1^ OGs with DP from 2 to 10 was used in the experiments. The formulation of Planticine^®^ was obtained by enzymatic hydrolysis of citrus pectin and was developed as a result of project number POIR.01.01.01-00-0024/15.

The experiment started on 01.12.2019 when seeds of *Solanum lycopersicum* L. cv. ‘Julia F1’ were sown in the pots (110 × 110 × 120 mm) containing a peat substrate. The pots were placed on growing benches with an area of 1.92 m^2^. The experiment included two combinations: untreated control (plants sprayed with distilled water) and plants sprayed with Planticine^®^. The experiment was conducted in a completely randomized design with three repetitions per treatment. The repetition was composed of 6 tomato plants. In the chamber, the plants were illuminated with 600 W light-emitting diodes (LEDs) (Fiona Lighting 300 LED, Senmatic A/S, Søndersø, Denmark). Two LEDs were placed above one growing bench. Photosynthetic photon flux density (PPFD) reaching the plants was approximately 200 µmol m^−2^·s^−1^, maintaining a photoperiod of 14 h of light and 10 h of darkness. The maximum air temperature was 25 °C during the day and 18 °C at night, and relative humidity was 60–65%.

The first foliar application of Planticine^®^ in a dose of 2 l ha^−1^ (concentration of Planticine^®^ in working solution 0.33%) was performed in the tomato growth phase of BBCH 14–16 (3 January 2020). The next two sprayings were performed every 5 days. After 48 h of the 3rd application of Planticine^®^, tomato leaves were collected for molecular and chemical analysis.

#### 4.1.1. RNA Extraction and RNA-Seq Analysis

For RNA-seq analysis, the last fully expanded, newly emerged leaves of control plants and plants treated with Planticine^®^ were collected and frozen immediately in liquid nitrogen and then stored at −80 °C until RNA extraction. Total RNA isolation was performed with NucleoSpin^®^ RNA (Macherey-Nagel, Düren, Germany) as described by the manufacturers. DNA contaminations were removed with the Turbo DNA-free kit (Thermo Fisher Scientific; Ambion; Austin, TX, USA) following the producer protocol. The quality and quantity of RNA were determined using NanoDrop 2000c (Thermo Fisher Scientific, Waltham, MA, USA) and gel electrophoresis under denaturing conditions. Three biological replicates were prepared, whereas each of them pooled RNA (in equal concentrations) obtained from 6 independent plants. The A260/A280 ratio and RNA integrity number (RIN) of each biological repetition were determined by a Bioanalyzer 2100 (Agilent 2100 Bioanalyzer; Agilent Technologies, Palo Alto, Santa Clara, CA, USA). Nine cDNA libraries prepared using the NEBNext^®^ UltraTM RNA Library Kit (Illumina, San Diego, CA, USA) were subjected to sequencing in PE150 (paired-ends mode, with 150 bp read length) on an Illumina HiSeq4000 (Illumina, San Diego, CA, USA). The RNA-Seq datasets generated for this study are deposited in the NCBI under BioProject PRJNA906914.

The raw sequences were subjected to adaptor removal using Cutadapt ver. 1.9.1 (http://cutadapt.readthedocs.io, accessed on 20 September 2021) and quality trimming and control using BBMap toolkit ver. 37.02 (https://jgi.doe.gov/data-and-tools/bbtools, accessed on 20 September 2021) and FASTQC ver. 0.11.5, (https://www.bioinformatics.babraham.ac.uk/projects/fastqc/, accessed on 20 September 2021), respectively. The quality filter was as follows: a Phred score (Q) = 20, minimal read length = 25 bp, and all unpaired reads were excluded. The high-quality reads were aligned to the *Solanum lycopersicum* reference genome (NCBI accession GCA_000188115.3) using Hisat2 package ver. 2.2.0 (http://daehwankimlab.github.io/hisat2, accessed on 15 October 2021) with extra parameters -dta --rna-strandness RF --novel-splicesite-outfile. Read counts were calculated using HTseq with the -s reverse parameter [55]. We applied DESeq2 ver. 1.18 (https://bioconductor.org/packages/release/bioc/html/DESeq2.html, accessed on 10 November 2021) to normalize with library size the gene expression levels and perform differential expression genes (DEGs) analysis by comparing the normalized read counts for a given gene between Planticine^®^-treated and control samples. Genes with a threshold of adjusted *p*-value/False Discovery Rate (FDR) ≤ 0.05 were considered to be differentially expressed. The GO (Gene Ontology) category enrichment analysis for DEGs was performed using topGO R/Bioconductor package ver. 2.38.1 [56]. The significance of occurrence for a certain GO term was determined using Fisher’s exact test (*p*-values ≤ 0.01) in combination with the “classic” and “elim” algorithms to test for GO-term overrepresentation within the three domains: biological process (BP), molecular function (MF) and cellular component (CC). For the KEGG pathway enrichment of the DEGs [57], we used the R package clusterProfiler ver. 3.6.0 tool (http://www.bioconductor.org/packages/release/bioc/html/clusterProfiler.html, accessed on 20 January 2022) with a *p*-value ≤ 0.05 as the cut-off criterion.

#### 4.1.2. SA, JA, and ABA Assays

In the samples of leaves, an analysis of the content of SA, JA, and ABA was performed. Three biological replicates were collected from control plants and plants treated with Planticine^®^. Each biological replicate sample consisted of the last fully developed leaves collected from 6 tomato plants. Extraction and determination of SA, JA, and ABA by the LC-MS/MS technique were performed according to the method described by Halka et al., 2019 [58] with the modification by Smoleń et al., 2020 [59]. The measurements were made using an HPLC Ultimate 3000 (Thermo Scientific, Germering, Germany) and spectrometer LC-MS/MS: 4500 Qtrap (Sciex, Framingham, MA, USA). Chromatographic separation was carried out on a Luna 3 µm phenyl-hexyl 100 Å column (Phenomenex, Torrance, CA, USA). Electrospray ionization in negative ion mode was used. MS/MS was performed for quantitative analysis. The LC-MS/MS system was controlled using Analyst 1.7 with HotFix 3 software, which was also used for data processing.

### 4.2. Greenhouse Trials with Powdery Mildew and Thrips on Tomatoes

In the scope of the studies, two independent trials were conducted to assess the efficacy of Planticine^®^ against biotic stress caused by powdery mildew and thrips in tomato cultivation. Planticine^®^ was applied in two doses of 2 l ha^−1^ and 3 l ha^−1^, which correspond to concentrations in a working solution of 0.33% and 0.5%, respectively. The number of working solutions was 600 l ha^−1^. Additionally, an adjuvant Silwet Gold (UPL, Warsaw, Poland) in a concentration of 0.015% was added to each spraying treatment. Untreated control and a chemical reference product (Scorpion 325 SC for powdery mildew and Mospilan 20 SP for thrips) were included in the experiments, which had a randomized complete block design with four repetitions per treatment. The plot size and the number of plants on each plot were 2.5 m^2^, 10 plants and 6 m^2^, 20 tomato plants in the experiment with powdery mildew and thrips, respectively. The efficacy evaluations of Planticine^®^ were performed according to the European and Mediterranean Plant Protection Organization (EPPO) guidelines, which define the standard procedures for the evaluation of plant protection products. The trials were performed according to EPPO guidelines PP 1/57(3) for powdery mildew and PP 1/160(2) for thrips.

#### 4.2.1. Trial with Powdery Mildew on Tomatoes

Studies on the efficacy of Planticine^®^ in the protection of the tomato cv. ‘Julia F1’ against powdery mildew (*Oidium neolycopersici*) was conducted in the greenhouse of the National Institute of Horticultural Research in Skierniewice. As elicitors primarily have a preventive function, the application of the tested product was made prophylactically before the potential outbreak of the disease occurred. Five Planticine^®^ sprayings were performed between 28 May and 6 July with a 7–10 day interval. The first application was in the growth phase of BBCH 53–61. The degree of infestation of tomato leaves by powdery mildew was assessed on 30 randomly selected leaves per plot in each repetition at 10 days after the 3rd (T3 + 10), 4th (T4 + 10), and 5th (T5 + 10) Planticine^®^ application. Disease ratings were based on a percentage of infestation of the leaf area on a scale of 0 to 8 (0—0% with no symptoms; 1—1% of leaf area with disease symptoms; 2—2–5% of the infested area; 3—6–15%; 4—16–25%; 5—26–50%; 6—51–75%; 7—76–100%). The efficacy of the tested products in the reduction of infestation by powdery mildew was calculated using the Abbott formula (Equation (1)).

Equation (1):(1)$$Efficacy\left(\%\right)=\frac{C-T}{C}\times 100,$$
where C = mean infestation level in the untreated control plots and T = mean infestation level in the treated plots.

#### 4.2.2. Trial with Thrips on Tomato

The efficacy trial on the tomato cv. ‘Manistella F1’ with thrips was carried out in the year 2020 in a greenhouse owned by Szymanowice Poland by the NEFSCIENCE company. The infestation of plants by *Frankliniella occidentalis* was natural. Plants were cultivated in a coco-peat substrate with a fertilization system. To determine the direct effects on adult thrips, the tested products were applied twice at a 7-day interval. The first spraying was in the growth phase of BBCH 72 on 17 September. Observations of the number of thrips on the leaves were conducted. From each plot in the experiment, 10 leaves were collected, and thrips were counted. The assessment of the number of thrips was performed before the treatments (T0), then 3 and 7 days after the 1st treatment (T1 + 3; T1 + 7), 7 days after the 2nd treatment (T2 + 7), and then every 7 days for the next 2 weeks (T2 + 14; T2 + 21). The efficacy of the formulations in the protection of tomatoes against *F. occidentalis* was calculated according to the Henderson–Tilton formula (Equation (2)).

Equation (2):(2)$$Efficacy\left(\%\right)=(1-\frac{n\;in\;C\;before\;treatment\times n\;in\;T\;after\;treatment}{n\;in\;C\;after\;treatment\times n\;in\;T\;before\;treatment})\times 100,$$
where: n—mean number of thrips from 4 repetitions, T—treated plots, and C—control plots.

### 4.3. Statistical Analysis

In the experiments with powdery mildew and thrips, the significance of differences between mean values was determined by one-way analysis of variance; Duncan’s multiple range tests were used to compare the means. Student’s *t*-test was used to determine statistically significant differences between the plants treated and untreated with Planticine^®^ grown in a climatic chamber. The analyses were conducted using Statgraphics Centurion software version 17.2.02 (64-bit) (Statpoint Technologies, Inc, Gambit CoiS, Cracow, Poland).

## 5. Conclusions

We reported a transcriptome analysis that includes data on the tomato’s response to treatment with Planticine^®^, a natural plant defense biostimulant based on oligomers of α(1→4)-linked D-galacturonic acids. The study provides evidence at the transcriptomic level for the positive effects of the foliar application of Planticine^®^ to biotic stresses. Analysis of differentially expressed genes (DEGs) revealed their involvement, in particular in the plant-pathogen interaction, plant hormone signal transduction, and MAPK signaling pathways. Moreover, our results proved the efficacy of its use as a natural plant resistance inducer in greenhouse conditions, especially against powdery mildew (*Oidium neolycopersici*) and thrips (*Frankliniella occidentalis*). The advantage of the use of Planticine^®^ containing natural substances over chemical plant protection products is the production of food that is free from pesticide residues and the reduction of environmental pollution. In addition, agrophages cannot develop immunity to inducers, as is the case with resistance developed toward active substances present in pesticides [60], and activated resistance refers to a broad spectrum of pathogens [61], as was shown in this study.

## Figures and Tables

**Figure 1 ijms-24-06494-f001:**
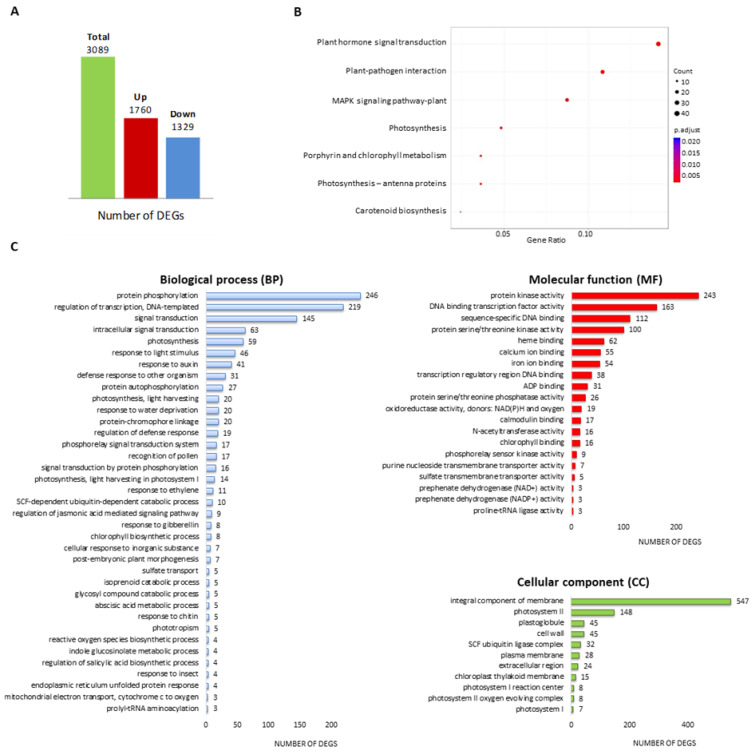
The expression patterns of Planticine^®^-responsive genes in tomato leaves. (**A**) Number of differentially expressed genes (DEGs), up- and down-regulated DEGs were marked in red and blue color, respectively; (**B**) KEGG pathway enrichment of DEGs; (**C**) GO enrichment of DEGs in three main categories: biological process (BP), molecular function (MF), and cellular component (CC); the X-axis indicates the number of genes, and Y-axis indicates the GO terms.

**Figure 2 ijms-24-06494-f002:**
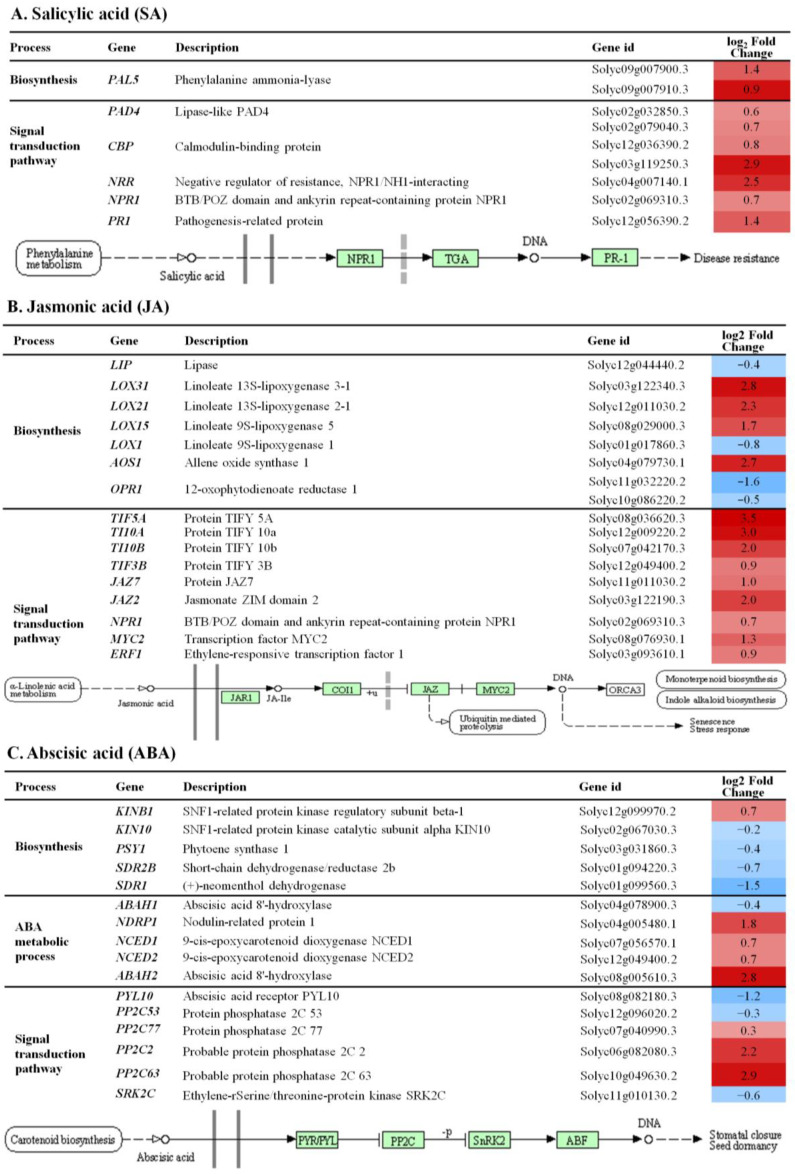
Differentially expressed genes (DEGs) in tomato leaves under Planticine^®^ treatment involved in biosynthesis and signal transduction pathway of salicylic acid (SA) (**A**), jasmonic acid (JA) (**B**) and abscisic acid (ABA) (**C**). Red and blue colors indicate up- and down-regulation of genes in the pathway, respectively. DEGs involved in plant hormone signal transduction pathway, based on the Kyoto Encyclopedia of Genes and Genomes (KEGG) pathway system, were indicated in green boxes.

**Table 1 ijms-24-06494-t001:** Resistance genes, including regulatory (ELLR, ERR, MAPK, TF) and resistance-related protein (RRP) and resistance-related metabolite (RRM) biosynthesis genes, differentially expressed in tomato leaves under Planticine^®^ treatment.

Gene Name	Description	Gene ID	Chromosome	log_2_ Fold Change Planticine^®^ vs. Control	False Discovery Rate (FDR)
**Elicitor and effector recognition receptor genes (ELRR and ERR)**					
*EIX1*	Receptor-like protein EIX1	Solyc07g008620.1	7	1.1	3 × 10^−4^
*EIX2*	Receptor-like protein EIX2	Solyc12g005610.2	12	1.0	4 × 10^−2^
*ELF*	Elongation factor Tu	Solyc03g112150.1	3	−0.6	1 × 10^−11^
*CERK1* (*LYK4*)	Chitin elicitor receptor kinase 1, LysM domain receptor-like kinase 4	Solyc02g089920.2	2	2.1	5 × 10^−4^
*CERK1* (*LYK3*)	Chitin elicitor receptor kinase 1, LysM domain receptor-like kinase 3	Solyc06g075030.1	6	−0.7	8 × 10^−3^
*CERK1*	Chitin elicitor receptor kinase 1	Solyc07g049180.3	7	0.8	1 × 10^−4^
RLK5	Receptor-like protein kinase 5	Solyc08g066310.2	8	1.0	1 × 10^−9^
RLK7	Receptor-like protein kinase 7	Solyc01g106500.3	1	1.0	3 × 10^−2^
*RLK*	Probable receptor-like protein kinase At1g33260	Solyc12g005450.1	12	3.4	1 × 10^−27^
	Putative receptor-like protein kinase At1g72540	Solyc06g062920.3	6	3.5	6 × 10^−5^
*CNGC*	Probable cyclic nucleotide-gated ion channel	Solyc05g050380.3	5	1.0	6 × 10^−4^
		Solyc03g098210.3	3	0.6	2 × 10^−3^
*FLII*	Flagellum-specific ATP synthase	Solyc01g107740.3	1	2.0	8 × 10^−21^
*FLS2*	LRR receptor-like serine/threonine-protein kinase FLS2	Solyc02g070920.3	2	−0.6	3 × 10^−6^
*SIK1*	LRR receptor-like serine/threonine-protein kinase SIK1	Solyc08g066320.3	8	3.2	6 × 10^−16^
*GSO2*	LRR receptor-like serine/threonine-protein kinase GSO2	Solyc10g052880.1	10	2.0	3 × 10^−7^
*RFK1*	Probable LRR receptor-like serine/threonine-protein kinase RFK1	Solyc02g071870.3	2	0.6	2 × 10^−2^
*FEI2*	LRR receptor-like serine/threonine-protein kinase FEI2	Solyc03g059490.1	3	0.6	3 × 10^−6^
*WAK*	Wall-associated receptor kinase-like	Solyc02g090110.3	2	1.2	5 × 10^−9^
*LRL*	Leaf rust 10 disease-resistance locus receptor-like protein kinase-like	Solyc01g008500.3	1	1.2	10 × 10^−5^
*SERK1* (*BAKIBKK1*)	Somatic embryogenesis receptor kinase 1 (brassinosteroid insensitive 1-associated receptor kinase 1)	Solyc04g072560.3	4	0.5	9 × 10^−4^
**Mitogen-activated protein kinases (MAPKs)**					
*SlMAPKKK76*	Calcium-dependent protein kinase 76	Solyc10g079130.2	10	1.3	7 × 10^−8^
*CDPK*	Calcium-dependent protein kinase	Solyc02g083850.3	2	1.0	2 × 10^−2^
*CDPK2*	Calcium-dependent protein kinase 2	Solyc03g033540.3	3	0.7	2 × 10^−3^
*CDPK4*	Calcium-dependent protein kinase 4	Solyc06g065380.3	6	−0.8	2 × 10^−4^
*CML*	Calmodulin-like protein (putative calcium-binding protein CML19)	Solyc02g094000.1	2	3.4	6 × 10^−4^
	Calmodulin-like protein (probable calcium-binding protein CML45	Solyc06g069740.1	6	2.1	2 × 10^−10^
	Calcium-binding protein CML37	Solyc11g071760.2	11	3.1	3 × 10^−3^
	Calmodulin-like protein (probable calcium-binding protein CML)	Solyc03g005040.1	3	3.0	2 × 10^−7^
	Calmodulin-like protein (probable calcium-binding protein CML)	Solyc06g073830.1	6	2.3	1 × 10^−8^
	Probable calcium-binding protein CML10	Solyc01g091465.1	1	1.8	3 × 10^−13^
	Calmodulin (calcium-binding protein CML24)	Solyc02g091500.1	2	1.6	2 × 10^−7^
	Calmodulin-like protein (probable calcium-binding protein CML46)	Solyc03g115930.2	3	1.4	5 × 10^−7^
	Calmodulin (calcium-binding protein CML23)	Solyc02g063350.1	2	1.0	6 × 10^−16^
	Calmodulin-like protein (probable calcium-binding protein CML45)	Solyc02g088090.1	2	1.0	2 × 10^−3^
	Probable calcium-binding protein CML36	Solyc10g079755.1	10	−0.4	1 × 10^−5^
*MPK3*	Mitogen-activated protein kinase 3	Solyc06g005170.3	6	1.9	3 × 10^−3^
*MKS1*	MAP kinase substrate 1	Solyc11g005720.1	11	1.4	4 × 10^−5^
**Transcription factor genes (TF) and other regulatory genes**					
*WRKY*	Probable WRKY 53	Solyc08g008280.3	8	3.4	9 × 10^−5^
	WRKY 22	Solyc10g011910.3	10	3.2	5 × 10^−4^
	Probable WRKY 41	Solyc03g007380.2	3	3.1	6 × 10^−12^
	Probable WRKY 40	Solyc03g116890.3	3	3.1	7 × 10^−5^
	Probable WRKY 60	Solyc08g067360.3	8	2.6	4 × 10^−9^
	WRKY 72A	Solyc03g113120.3	3	2.6	1 × 10^−2^
	WRKY 6	Solyc02g080890.3	2	2.5	3 × 10^−9^
	Probable WRKY 33	Solyc09g014990.3	9	2.4	2 × 10^−2^
	WRKY 70	Solyc03g095770.3	3	2.3	6 × 10^−3^
	Probable WRKY 30	Solyc10g009550.3	10	2.1	6 × 10^−31^
	Probable WRKY 7	Solyc04g078550.3	4	1.9	2 × 10^−11^
	Probable WRKY 75	Solyc05g015850.3	5	1.7	4 × 10^−4^
	Probable WRKY 51	Solyc04g051690.3	4	1.5	7 × 10^−5^
	Probable WRKY 11	Solyc12g096350.2	12	1.5	2 × 10^−6^
	WRKY WRKY76	Solyc06g068460.3	6	1.4	4 × 10^−2^
	Probable WRKY 50	Solyc08g062490.3	8	1.4	1 × 10^−5^
	WRKY 22	Solyc01g095100.3	1	1.2	3 × 10^−11^
	Probable WRKY 19	Solyc01g104910.3	1	−0.3	3 × 10^−3^
	Probable WRKY 21	Solyc09g066010.3	9	−0.4	4 × 10^−2^
*ERF*	Ethylene-responsive TF ERF22	Solyc11g042560.1	11	5.4	2 × 10^−7^
	Ethylene-responsive TF ERF112	Solyc02g090770.1	2	4.5	2 × 10^−4^
	Ethylene-responsive TF ERF109	Solyc10g050970.1	10	4.2	2 × 10^−3^
	Ethylene-responsive TF ERF17	Solyc06g054630.2	6	3.8	7 × 10^−15^
	Ethylene-responsive TF ERF25	Solyc06g035700.1	6	3.2	1 × 10^−17^
	Ethylene-responsive TF ERF109	Solyc01g108240.3	1	2.9	1 × 10^−2^
	Ethylene-responsive TF ERF118	Solyc04g009435.1	4	2.8	1 × 10^−4^
	Ethylene-responsive TF ERF12	Solyc11g012980.1	11	2.1	3 × 10^−6^
	Ethylene-responsive TF ERF5	Solyc08g078190.1	8	1.5	2 × 10^−4^
	Ethylene-responsive TF ERF4	Solyc07g053740.1	7	1.0	2 × 10^−2^
	Ethylene-responsive TF ERFC3	Solyc04g014530.1	4	−1.4	4 × 10^−4^
	Ethylene-responsive TF ERF13	Solyc01g090340.2	1	−2.2	1 × 10^−16^
*MYB*	Transcription factor MYB14	Solyc07g054980.2	7	3.7	7 × 10^−3^
	Transcription factor MYB77	Solyc04g079360.1	4	3.2	3 × 10^−24^
	Transcription factor MYB14	Solyc12g005640.2	12	2.0	2 × 10^−3^
	Transcription factor MYB15	Solyc07g053230.3	7	1.9	2 × 10^−7^
	Transcription factor MYB102	Solyc02g079280.3	2	1.6	2 × 10^−2^
	Transcription factor MYB20	Solyc11g011050.2	11	1.5	4 × 10^−3^
	Transcription factor MYB73	Solyc04g078420.1	4	0.9	1 × 10^−9^
	Transcription factor MYB36	Solyc09g008250.3	9	−1.2	8 × 10^−3^
	Transcription factor MYB16	Solyc02g088190.3	2	−1.1	2 × 10^−8^
	Transcription factor MYB113	Solyc10g086250.2	10	−1.0	1 × 10^−7^
*NAC*	NAC domain-containing protein 90	Solyc11g068620.2	11	2.4	4 × 10^−3^
	NAC domain-containing protein 22	Solyc10g055760.2	10	1.2	1 × 10^−5^
	NAC domain-containing protein 79	Solyc03g115850.3	3	1.0	3 × 10^−3^
	NAC domain-containing protein 17	Solyc04g072220.3	4	0.4	4 × 10^−2^
	NAC domain-containing protein 35	Solyc01g102740.3	1	−0.5	2 × 10^−3^
*bZIP*	BZIP TF family protein expressed	Solyc01g110480.3	1	1.2	4 × 10^−9^
*WFI1*	whitefly-induced gp91 (circRNA)	Solyc03g117980.3	3	2.1	2 × 10^−4^
*RBOHC*	Respiratory burst oxidase homolog protein C	Solyc03g117980.3	3	2.1	2 × 10^−4^
**Resistance-related protein (RRP) biosynthetic genes**					
*A70*	Pathogen-associated molecular patterns-induced protein A70	Solyc01g079660.2	1	3.2	1 × 10^−3^
*PTI5*	Pathogenesis-related genes transcriptional activator PTI5	Solyc02g077370.1	2	2.1	9 × 10^−7^
*PTI6*	Pathogenesis-related genes transcriptional activator PTI6	Solyc06g082590.1	6	0.9	1 × 10^−2^
*PR1*	Thaumatin, pathogenesis-related protein	Solyc12g056390.2	12	1.4	8 × 10^−6^
*PRS2*	Pathogenesis-related protein STH-2	Solyc12g096960.2	12	1.2	2 × 10^−3^
*E70*	Exocyst complex protein EXO70	Solyc09g005830.1	9	2.0	2 × 10^−2^
		Solyc06g075610.1	6	1.4	2 × 10^−4^
		Solyc11g073010.1	11	1.0	3 × 10^−2^
*RIN4*	RPM1-interacting protein 4	Solyc09g059430.3	9	1.4	1 × 10^−6^
		Solyc06g083390.3	6	1.2	1 × 10^−6^
*RIN1*	RPM1-interacting protein 1	Solyc11g010170.2	11	0.9	2 × 10^−6^
*R13L4*	Disease resistance RPP13-like protein 4	Solyc02g084890.2	2	2.0	2 × 10^−3^
*PBS1*	Probable serine/threonine-protein kinase PBL7	Solyc11g072660.2	11	1.4	1 × 10^−5^
	Probable serine/threonine-protein kinase PBL19	Solyc08g077560.3	8	2.3	3 × 10^−9^
	Probable serine/threonine-protein kinase PBL3	Solyc01g010660.3	1	1.8	4 × 10^−9^
*HSP71*	Heat shock cognate 70 kDa protein 1	Solyc06g076020.3	6	2.4	3 × 10^−2^
*HSP12*	18.2 kDa class I heat shock protein	Solyc10g086680.1	10	1.9	4 × 10^−6^
*HSF24*	Heat shock factor protein HSF24	Solyc02g090820.3	2	1.5	2 × 10^−8^
*HS704*	Heat shock 70 kDa protein 4	Solyc06g005435.1	6	−1.8	1 × 10^−3^
*EDS1*	Protein EDS1	Solyc06g071280.3	6	0.8	2 × 10^−5^
*KCS11*	3-ketoacyl-CoA synthase 11	Solyc06g065560.2	6	1.8	9 × 10^−13^
		Solyc09g065800.3	9	−2.2	1 × 10^−10^
*HIR1*	Hypersensitive-induced reaction 1 protein	Solyc03g113220.3	3	0.4	3 × 10^−2^
*ROQ1*	Disease resistance protein Roq1	Solyc01g102840.3	1	2.8	1 × 10^−8^
*RPV1*	Disease resistance protein RPV1	Solyc01g102880.2	1	1.3	1 × 10^−8^
*RUN1*	Disease resistance protein RUN1	Solyc04g007320.2	4	1.2	1 × 10^−5^
*NGR1*	Probable disease resistance protein	Solyc02g090380.3	2	1.1	4 × 10^−7^
*EDR4*	Protein enhanced disease resistance 4	Solyc03g095610.3	3	0.6	3 × 10^−2^
*DGK1*	Diacylglycerol kinase 1	Solyc03g115370.3	3	0.5	6 × 10^−3^
*MCA1*	Metacaspase-1	Solyc03g094160.3	3	1.5	3 × 10^−7^
**Resistance-related metabolite (RRM) biosynthetic genes**					
*LOX31*	Linoleate 13S-lipoxygenase 3-1	Solyc03g122340.3	3	2.8	1 × 10^−18^
*LOX21*	Linoleate 13S-lipoxygenase 2-1	Solyc12g011030.2	12	2.3	3 × 10^−26^
*LOX15*	Probable linoleate 9S-lipoxygenase 5	Solyc08g029000.3	8	1.7	2 × 10^−2^
*AOS*	Allene oxide synthase 1	Solyc04g079730.1	4	2.7	6 × 10^−3^
*TPS5*	(R)-linalool synthase TPS5	Solyc01g105890.3	1	1.1	1 × 10^−4^
*ACS3*	1-aminocyclopropane-1-carboxylate synthase 3	Solyc02g063540.2	2	3.7	1 × 10^−4^
		Solyc02g091990.3	2	2.8	6 × 10^−6^
*ACS CMW33*	1-aminocyclopropane-1-carboxylate synthase CMW33	Solyc08g081555.1	8	1.8	5 × 10^−9^
*ACS*	1-aminocyclopropane-1-carboxylate synthase	Solyc08g081540.3	8	1.7	3 × 10^−5^

**Table 2 ijms-24-06494-t002:** Content of salicylic acid (SA), jasmonic acid (JA), and abscisic acid (ABA) in tomato leaves.

Treatment	Dose L ha^−1^	µg kg^−1^ d.w.		
		SA	JA	ABA
Control	-	671.7 ± 52.8	8.39 ± 2.9	563.5 ± 24.2
Planticine^®^	2	885.8 ± 50.6	5.16 ± 1.16	585.8 ± 40.9
Test *t*-student	-	*	*	NS

The significance was declared at *p* ≤ 0.05; *—significant differences, Non-significant differences (NS).

**Table 3 ijms-24-06494-t003:** Average percentage of tomato leaves infestation by *Oidium neolycopersici* and efficacy in powdery mildew limitation depending on the application of Planticine^®^ and chemical standard.

Treatment	Dose per ha (L ha^−1^)	Observation					
		T3 + 10		T4 + 10		T5 + 10	
		Leaves Infestation (%)/Efficacy (%)					
Control	-	8.78 c	-	30.2 d	-	68.8 d	-
Planticine^®^	2	0.78 b	91%	7.4 b	76%	36.6 b	47%
Planticine^®^	3	0.84 b	90%	10.4 c	66%	42.9 c	38%
Scorpion 325 SC	1	0.0 a	100%	0.0 a	100%	0.2 a	99%

Observation: T3 + 10—10 days after the 3rd application, T4 + 10—10 days after the 4th application, T5 + 10—10 days after the 5th application; means followed by the same letters are not significantly different for *p* ≤ 0.05.

**Table 4 ijms-24-06494-t004:** Number of adults of *Frankliniella occidentalis* on tomato leaves and efficacy of their reduction depending on the application of Planticine^®^ and chemical standard.

Treatment	Dose per ha (L/kg ha^−1^)	Observation										
		T0	T1 + 3		T1 + 7		T2 + 7		T2 + 14		T2 + 21	
		Number of Adults/Efficacy (%)										
Control	-	3.5 a	1.75 a	-	2.25 a	-	2.75 ab	-	2.75 ab	-	5.0 a	-
Planticine^®^	2	7.25 a	2.0 a	45	3.75 a	20	2.25 a	61	1.5 a	74	3.25 a	69
Planticine^®^	3	2.25 a	3.25 a	-	3.25 a	-	3.25 ab	-	6.0 c	-	1.0 a	69
Mospilan 20 SP	0.24	6.0 a	6.5 b	-	5.25 a	-	5.25 b	-	3.5 b	26	2.0 a	77

Observation: T0—before application, T1 + 3, T1 + 7—3, and 7 days after 1st application., T2 + 7, T2 + 14, T2 + 21—7, 14, and 21 days after the 2nd application. Means followed by the same letters are not significantly different for *p* ≤ 0.05.

## Data Availability

Data are contained within the article or Appendix A. The RNA-Seq datasets generated for this study are deposited in the NCBI under BioProject PRJNA906914.

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
