# Peer review of "Transcriptome Dynamics Underlying Planticine®-Induced Defense Responses of Tomato (Solanum lycopersicum L.) to Biotic Stresses"

_ijms, 2023, doi:10.3390/ijms24076494_

Round 1

Reviewer 1 Report

The authors provide a study on the molecular basis of biological activity of Planticine and its efficacy of its use as a natural plant resistance inducer and its efficacy in greenhouse condition. It is an interesting work. However, there are some issues as follows should be addressed.

1. The title of the manuscript is 'Transcriptome dynamics...', yet neither the generated data nor the analysis is related with dynamics, which can be a bit confusing.

2. As a result of this study, 3049 genes are detected. But the number of the genes is too large for a scientific report. Therefore, it is suggested that further analysis be conducted to reduce the size o this gene module.

3. The study's focus on differential expression analysis and GO analysis alone may be considered too simplistic. Therefore, it is recommended that more specialized analyses be conducted.

4. It is recommended to include a figure that briefly outlines the regulatory mechanism for natural plant resistance, highlighting key genes involved in the process.

Author Response

Dear Reviewer,

First of all, we would like to thank you for the evaluation of our manuscript. Thank you for your time and effort put into the assessment of the publication. While improving the manuscript we took into account suggestions proposed by you.

The manuscript has been linguistically checked by a English native speaker. All revisions to the manuscript were marked up using the “Track Changes” and yellow color.

REVIEWER 1

  1. The title of the manuscript is 'Transcriptome dynamics...', yet neither the generated data nor the analysis is related with dynamics, which can be a bit confusing.

Response: The title has been changed because it did not fully reflect the meaning of the work.

The title: Transcriptome dynamics underlying Planticine®-induced defense responses of tomato (Solanum lycopersicum L.) to biotic stresses.

  1. As a result of this study, 3049 genes are detected. But the number of the genes is too large for a scientific report. Therefore, it is suggested that further analysis be conducted to reduce the size of this gene module.

Response: In our work, we wanted to identify all genes that were characterized by differential expression profile, particularly those related to the mechanism: Plant pathogen interaction. Reducing the gene pool would be associated with the selection of a lower value of a threshold of adjusted p value / False Discovery Rate (FDR) than 0.05. At the first stage of our work with the RNA-seq data, we compared DEGs at p values of 0.01 and 0.001. We consciously decided to select DEGs pools with p value (FDR) ≤0.05 for further analysis, because thanks to this approach we were able to identify most genes from the Plant pathogen interaction pathway, as presented in Table 1, where each gene is assigned an FDR. For further detailed analyses, we will rely on the reduced pool of identified genes.

  1. The study's focus on differential expression analysis and GO analysis alone may be considered too simplistic. Therefore, it is recommended that more specialized analysis be conducted.

Response: Thanks to RNA-seq analysis, we were able to identify transcripts related to the response to the biostimulant Planticine®, thanks to which we identified biological processes and pathways. Further studies will be related to the analysis of gene expression after treatment with Planticine and after treatment with thrips, powdery mildew and other agrophages. Currently, we conduct microscopic imaging, as well as biochemical analyzes related to the determination of the profile of phytohormones and selected metabolites.

  1. It is recommended to include a figure that briefly outlines the regulatory mechanism for natural plant resistance, highlighting key genes involved in the process.

Response:  Thank you for this remark. We agree with you that the summary is needed in the manuscript. We added chapter 4 – Conclusion, in which we did the summary of the key findings and their implications.

Reviewer 2 Report

A brief summary 

The molecular basis of the biological activity of Planticine® (an oligomers of α(1→4)-linked D-galacturonic acids from 2 to 10 in length) and its effectiveness as a natural inducer of plant resistance to pests was investigated. RNA sequencing methods showed that 1760 tomato genes were activated as a result of exposure to Planticine®, and 1329 genes were suppressed. Planticine® affected plant-pathogen interaction, phytohormone signal transduction, MAPK signaling, photosynthesis, and transcriptional regulation.

Broad comments 

The article is of considerable interest both from the point of view of the practical application of Planticine® and from the point of view of studying the molecular biological processes of plant interaction with pathogens.

Specific comments 

Line 125: first give a detailed name and then its abbreviation in brackets: (KEGG)

Line 131: first give a detailed name and then its abbreviation in brackets: (GO)

Table 1: FDR - reveal abbreviation

and so on

The text contains a large number of abbreviations used in this field of science. But the IJMS journal is not a highly specialized journal, so it is desirable that all abbreviations be transcribed at the place of their first appearance.

Author Response

Dear Reviewer,

First of all, we would like to thank you for the evaluation of our manuscript. Thank you for your time and effort put into the assessment of the publication. While improving the manuscript we took into account suggestions proposed by you.

The manuscript has been linguistically checked by a English native speaker. All revisions to the manuscript were marked up using the “Track Changes” and yellow color.

Specific comments 

Line 125: first give a detailed name and then its abbreviation in brackets: (KEGG)

Response: It has been changed

Line 131: first give a detailed name and then its abbreviation in brackets: (GO)

Response: It has been changed

Table 1: FDR - reveal abbreviation

Response: It has been changed

and so on…

The text contains a large number of abbreviations used in this field of science. But the IJMS journal is not a highly specialized journal, so it is desirable that all abbreviations be transcribed at the place of their first appearance.

Response: We went through the text carefully and we added full explanation of the abbreviations. Thank you very much for this remark.

Reviewer 3 Report

ijms-2291273-peer-review-v1

The manuscript ‘Transcriptome dynamics underlying elicitor-induced defense responses of tomato (Solanum lycopersicum L.)’ by Rakoczy-Lelek et al. is well-conceptualized, executed, and presented. I have a few points the author could address to improve the manuscript.

In the title, the defense response ‘against the stress is missing’. The authors may include the biotic stresses ‘powdery mildew and thrips’. The title may be revised as ‘Transcriptome dynamics underlying elicitor-induced defense responses of tomato (Solanum lycopersicum L.) to biotic stresses’ 

Please mention some critical results in the abstract, such as the quantifiable (-folds) highly up and down-regulated genes of the transcriptional profile.

Line 52-53: Plant exhibits innate immunity in terms of ROS, HR, Callose deposition, lignin peroxidation etc. what is innate resistance????

Line 107-108: Please elaborate on the aim of the study.

Results are well-presented.

The qPCR validation of top-up and down-regulated genes is missing, which is required to validate RNASeq data.

Please re-write the results of ‘2.3. Powdery mildew on tomato leaves’ and ‘2.4. Thrips on tomato leaves’

Tables: Please use decimals ‘.’ Instead of ‘,’ while presenting the data.

Line 251: Please use the term ‘Non-significant (NS)’.

Why have the authors not performed the RNASeq of powdery mildew and thrips-infested leaves before and after application of Planticine/ please justify.

The MS should contain a conclusion section with the key findings and their implications.

Line 468-469: ‘and was developed as a result of the project number POIR.01.01.01-00-0024/15’…may be deleted.

Line 482-485: Please revise.

Line 487: ‘the last fully expanded leaves’…The newly emerged or the older leaves? Please clarify.

Line 503-524: Please mention the date of access for the software used for bioinformatics analysis. Please write ‘p-value ≤ 0.05’ in italics.

Line 532: ‘p-value ≤ 0.05’, please mention the city and country.

Line 563: ‘3-rd’…..Please make rd, th, superscript.

Line 583-585: Please derive the equation using MS equation tool and number it correctly.

References may be arranged as per the journal pattern. Please cross-check the references cited with the list.

Although the overall grammar score is satisfactory, the authors are requested to polish the language, punctuation, and grammar once again.

I recommend the MS for acceptance with minor corrections.

Good luck with the revision.

Author Response

Dear Reviewer,

First of all, we would like to thank you for the evaluation of our manuscript. Thank you for your time and effort put into the assessment of the publication. While improving the manuscript we took into account suggestions proposed by you.

The manuscript has been linguistically checked by a English native speaker. All revisions to the manuscript were marked up using the “Track Changes” and yellow color.

REVIEWER 3

  1. In the title, the defense response ‘against the stress is missing’. The authors may include the biotic stresses ‘powdery mildew and thrips’. The title may be revised as ‘Transcriptome dynamics underlying elicitor-induced defense responses of tomato (Solanum lycopersicum) to biotic stresses’ 

Response: The title has been changed. Thank you for your valuable remark.

The title: Transcriptome dynamics underlying Planticine®-induced defense responses of tomato (Solanum lycopersicum L.) to biotic stresses.

  1. Please mention some critical results in the abstract, such as the quantifiable (-folds) highly up and down-regulated genes of the transcriptional profile.

Response: We added information about the key identified genes.

  1. Line 52-53: Plant exhibits innate immunity in terms of ROS, HR, Callose deposition, lignin peroxidation etc. what is innate resistance????

Response: Innate resistance has been change to Innate immunity.

Plants respond to infection using a two-branched innate immune system. The first branch recognizes and responds to molecules common to many classes of microbes, including non-pathogens. The second responds to pathogen virulence factors, either directly or through their effects on host targets.

  1. Line 107-108: Please elaborate on the aim of the study.

Response: The aim of this study, was to determine the efficacy of Planticine® in reducing damage caused by agrophages and determining mode of action this biostimulant. The transcriptome analysis of a tomato treated with Planticine® proved a good method and confirmed the field observations that Planticine® works as an elicitor.

  1. The qPCR validation of top-up and down-regulated genes is missing, which is required to validate RNASeq data.

Response: We agree that it will be worth to validate DEGs on plants from independent experiment, to confirm that changes that we observed are reproducible in subsequent years or under slightly different conditions. We are aware of the enormous variability of transcriptomes as a result of sometimes slight changes in the external environment. However, we are convinced that the experimental setup that we used and the sequencing of a bulk sample RNA extracted from individual plants sufficiently eliminate the signal of the genetic background and variation resulting from the growing conditions, allowing to reliably capture the real differences resulting from the applied treatment. Therefore, we see no biological justification for validating the obtained result. Since until now the RNA-seq technology is still state to art, very sensitive and reliable, validation of the method itself is unnecessary (Fang and Cui, 2011, Briefings in bioinformatics, 12(3), 280-287;  Coenye, T. 2021; Biofilm, 3).

  1. Please re-write the results of ‘3. Powdery mildew on tomato leaves’ and ‘2.4. Thrips on tomato leaves’

Response: It has been changed.

  1. Tables: Please use decimals ‘.’ Instead of ‘,’ while presenting the data.

Response: It has been changed.

  1. Line 251: Please use the term ‘Non-significant (NS)’.

Response: It has been changed.

  1. Why have the authors not performed the RNASeq of powdery mildew and thrips-infested leaves before and after application of Planticine/ please justify.

Response: In the first stage, we decided to check what is the response at the transcriptome level after treating health plants with the biostimulant alone, to indicate what effect it may have. Further studies will be related to the analysis of gene expression after treatment with Planticine plants affected by the agrophages. Currently, we conduct microscopic imaging, as well as biochemical analyzes related to the determination of the profile of phytohormones and selected metabolites.

  1. The MS should contain a conclusion section with the key findings and their implications.

Response: Thank you for this remark. We added chapter 4 – Conclusion, in which we did the summary of the key findings and their implications. We agree that such summary is needed in the manuscript.

  1. Line 468-469: ‘and was developed as a result of the project number POIR.01.01.01-00-0024/15’…may be deleted.

Response: The number of the grant is needed to settle the project.

  1. Line 482-485: Please revise.

Response: The part of the text has been corrected. Thank you for your valuable remark.

  1. Line 487: ‘the last fully expanded leaves’…The newly emerged or the older leaves? Please clarify.

Response: It has been changed in the text. There were newly emerged.

  1. Line 503-524: Please mention the date of access for the software used for bioinformatics analysis. Please write ‘p-value ≤ 0.05’ in italics.

Response: It has been changed, it was added the accesses for the software.

  1. Line 532: ‘p-value ≤ 0.05’, please mention the city and country.

Response: It has been changed.

  1. Line 563: ‘3-rd’…..Please make rd, th, superscript.

Response: It has been changed.

  1. Line 583-585: Please derive the equation using MS equation tool and number it correctly.

Response: It has been changed.

  1. References may be arranged as per the journal pattern. Please cross-check the references cited with the list.

Response: It has been changed.

  1. Although the overall grammar score is satisfactory, the authors are requested to polish the language, punctuation, and grammar once again.

Response: The native English speaker reviewed the manuscript

Author Response

Dear Reviewer

First of all, we would like to thank you for the evaluation of our manuscript. Thank you for your time and effort put into the assessment of the publication. While improving the manuscript we took into account suggestions proposed by you.

The manuscript has been linguistically checked by a English native speaker. All revisions to the manuscript were marked up using the “Track Changes” and yellow color.

  1. The manuscript is not well written and needs to be reworked. A native English speaker needs to review the manuscript. However, the work offers information that would be found interesting to plant biologists.

Response: All co-authors read once again manuscript, we did some modification, some phrases are rewritten. Moreover, the manuscript has been linguistically checked by a English native speaker.

We are very grateful the Reviewer for appreciation of our work.

  1. Although RNA-Seq data are analyzed and organized to provide the overall effect of Planticine, DEGs are not confirmed by qRT-PCR. It is suggested to quantify some DEGs (up and down) to validate the RNA-Seq results.

Response: We agree that it will be worth to validate DEGs on plants from independent experiment, to confirm that changes that we observed are reproducible in subsequent years or under slightly different conditions. We are aware of the enormous variability of transcriptomes as a result of sometimes slight changes in the external environment. However, we are convinced that the experimental setup that we used and the sequencing of a bulk  sample RNA extracted from individual plants sufficiently eliminate the signal of the genetic background and variation resulting from the growing conditions, allowing to reliably capture the real differences resulting from the applied treatment.  therefore, we see no biological justification for validating the obtained result. Since until now the RNAseq technology is still state to art, very sensitive and reliable, validation of the method itself is unnecessary (Fang and Cui, 2011, Briefings in bioinformatics, 12(3), 280-287;  Coenye, T. 2021; Biofilm, 3).

  1. On Figure 1C, GO enrichment analysis should be carried out separately: up-regulated genes and down-regulated genes. Same for Table 1.

Response: Thank you for this comment. We agree with you that this approach would allow us to indicate pathways / processes that are activated / inhibited by Planticine®. You can find in the Supplementary data S2 Gene Ontology enrichment results in three categories (BP, MF and CC), which show that only in the photosynthesis process all differentially expressed genes are characterized by up-regulation, and in the remaining processes DEGs are both up- and downregulated. Therefore, in chapters 2.1.1 and 2.1.2, in which we looked at in detail all genes related to plant-pathogen interaction and SA, JA and ABA pathways. In the Table 1 you can find for each single DEG log2 Fold Change Planticine® vs. Control and thanks to that you can see which gene is up- and down-regulated.

  1. Citation to important works regarding plant immunity is lacking. For example, Peng et al. Convergent and Divergent Signaling in PAMP-Triggered Immunity and Effector-Triggered Immunity. Mol. Plant-Microbe Interact. MPMI 2018, 31, 403– 409; Zhou and Zhang. Plant Immunity: Danger Perception and Signaling. Cell 2020, 181, 978–989; Ngou et al. Thirty Years of Resistance: Zig-Zag through the Plant Immune System. Plant Cell 2022, 34, 1447–1478; Ramírez-Zavaleta et al. An Overview of PRR- and NLR-Mediated Immunities: Conserved Signaling Components across the Plant Kingdom That Communicate Both Pathways. Int. J.Mol. Sci. 2022, 23, 12974.

Response: The manuscript was enriched by citing prominent works in the field. Thank you for your valuable remark.

  1. Finally, the work of Gong et al. Cross-Microbial Protection via Priming a Conserved Immune Co-Receptor through Juxtamembrane Phosphorylation in Plants. Cell Host Microbe 2019, 26, 810–822 (and others) is not mentioned in the discussion. It is suggested to enrich the discussion by citing prominent works in the field.

Response: The discussion was enriched by citing prominent work in the field. Thank you for your valuable remark.

Minor observations: Note: Given that there are many sections that need to be rewritten, below is only covered the abstract, aimed to show that a native English speaker must review the manuscript.

Lines 2-3: I suggest to change the title as follows: “Transcriptome dynamics underlying Planticine-induced defense responses of tomato (Solanum lycopersicum L.)”

Response: It has been changed.

Lines 19-20: Such information is not clear, but the main idea should be: Induction of natural defense mechanisms in plants is considered to be one of the most important strategies used in integrated pest management (IPM).

Response: It has been changed.

Lines 20-21: It should be: Plant immune inducers could reduce the use of chemicals for plant protection and their harmful impacts on the environment. Line 22: It should be “based on oligomers”.

Response: It has been changed.

Line 23: Replace “is” with “are”. Line 23: It should be “The aim of this study was to…” (there should not be a comma after “study”).

Response: It has been changed.

Lines 23-24: It should be “the molecular basis of Planticine’s biological activity”.

Response: It has been changed.

Lines 24-25: The word “efficacy” is used twice and makes the sentence not clear at all. On the other hand, replace “greenhouses condition” to “greenhouse conditions”.

Response: It has been changed.

Lines 25-28: All this information is mixed up. Line 30: It should be “differentially expressed genes (DEGs)”. In summary, the work could be published if the above observations are attended.

Response: It has been changed.

Round 2

Reviewer 1 Report

There is no further question for the manuscript.